# Glycemic variability and all-cause mortality in a large prospective southern European cohort of patients with differences in glycemic status

**Miguel A. Salinero-Fort**[1,2,3,4,5]*, **F. Javier San Andrés-Rebollo**[1,6], **Juan Cárdenas-Valladolid**[1,2,5,7], **José M. Mostaza**[2,8], **Carlos Lahoz**[2,8], **Fernando Rodriguez-Artalejo**[2,9], **Paloma Gómez-Campelo**[2,10], **Pilar Vich-Pérez**[1,11], **Rodrigo Jiménez-García**[12], **Ana López de Andrés**[12], **José M. de Miguel-Yanes**[13], **on behalf the MADIABETES and SPREDIA Consortium**[¶]

1 Foundation for Research and Biomedical Innovation of Primary Care of the Community of Madrid (FIIBAP), Madrid, Spain, 2 The Hospital La Paz Institute for Health Research (IdiPAZ), Madrid, Spain, 3 Health Services and Chronic Conditions Research Network (REDISSEC), Madrid, Spain, 4 General Subdirectorate of Research and Documentation, Department of Health, Madrid, Spain, 5 Alfonso X El Sabio University, Madrid, Spain, 6 Las Calesas Health Center, Madrid, Spain, 7 Information Systems Department, Primary Health Care Management, Madrid, Spain, 8 Lipids and Vascular Risk Unit, Internal Medicine, University Hospital La Paz-Cantoblanco-Carlos III, Madrid, Spain, 9 Department of Preventive Medicine and Public health, Universidad Autónoma de Madrid-IdiPAZ, CIBERESP (CIBER of Epidemiology and Public Health), and IMDEA-Food Institute, CEI UAM+CSIC, Madrid, Spain, 10 Foundation for Biomedical Research of La Paz University Hospital (FIBHULP), Madrid, Spain, 11 Los Alpes Health Center, Madrid, Spain, 12 Department of Public Health & Maternal and Child Health, Faculty of Medicine, Universidad Complutense de Madrid, Madrid, Spain, 13 Internal Medicine Department, Gregorio Marañón General University Hospital, School of Medicine, Complutense University of Madrid, Gregorio Marañón Health Research Institute (IiSGM), Madrid, Spain

¶ Membership of MADIABETES and SPREDIA Consortium is provided in the Acknowledgments.
* miguel.salinero@salud.madrid.org

## Abstract

### Background

Few studies have analyzed the relationship between glucose variability (GV) and adverse health outcomes in patients with differences in glycemic status. The present study tests the hypothesis that GV predicts all-cause mortality regardless of glycemic status after simple adjustment (age and sex) and full adjustment (age, sex, cardiovascular disease, hypertension, use of aspirin, statins, GLP-1 receptor agonists, SGLT-2 inhibitors and DPP-4 inhibitors, baseline FPG and average HbA1c).

### Methods

Prospective cohort study with 795 normoglycemic patients, 233 patients with prediabetes, and 4,102 patients with type 2 diabetes. GV was measured using the coefficient of variation of fasting plasma glucose (CV-FPG) over 12 years of follow-up. The outcome measure was all-cause mortality.

### Results

A total of 1,223 patients (657 men, 566 women) died after a median of 9.8 years of follow-up, with an all-cause mortality rate of 23.35/1,000 person-years. In prediabetes or T2DM

**Data Availability Statement:** There are restrictions on the availability of data for the MADIABETES study, due to the signed consent agreements around data sharing, which only allow access to external researchers for research following the project purposes. Requestors wishing to access the MADIABETES data used in this study can request it to the MADIABETES Steering Committee: estudios.fiibap@salud.madrid.org The request will then be passed to members of the MADIABETES Steering Committee for deliberation. However, the minimal data set is available from the Institutional Repository of the Regional Health System of the Community of Madrid (https://repositoriosaludmadrid.es/)

**Funding:** This study was funded by Instituto de Salud Carlos III through projects "PI15/00259" and "PI18/01025" and co-funded by the European Regional Development Fund, "A way of shaping Europe". The funders had no role in study design, data collection and analysis, decision to publish, or preparation of the manuscript.

**Competing interests:** The authors have declared that no competing interests exist.

**Abbreviations:** ADVANCE, Action in Diabetes and Vascular Disease: Preterax and Diamicron Modified Release Controlled Evaluation; ALLHAT, Antihypertensive and Lipid Lowering Treatment to Prevent Heart Attack Trial; CI, confidence Interval; COVID-19, Coronavirus disease 2019; CV-FPG, coefficient of variation of fasting plasma glucose; DBP, diastolic blood pressure; DM, diabetes mellitus; EUROASPIRE, European Action on Secondary and Primary Prevention by Intervention to Reduce Events; FINDRISC, Finnish Diabetes Risk Score; FPG, fasting plasma glucose; GV, glucose variability; HbA1c, glycated hemoglobin; HR, hazard ratio; IGT, impaired glucose tolerance; INDEF, National Institute of Deaths; MAGE, mean amplitude of glycemic excursions; OGTT, standard oral glucose tolerance test; OR, odds ratio; PSM, propensity score matching; SBP, diastolic blood pressure; SD-FPG, standard deviation of fasting plasma glucose; SD-HbA1c, standard deviation of HbA1c; SPREDIA, Screening prediabetes and type 2 diabetes; SPSS, Statistical Package for the Social Sciences; T1DM, type 1 diabetes mellitus; T2DM, type 2 diabetes mellitus; WHO, World Health Organization.

patients, the fourth quartile of CV-FPG exerted a significant effect on all-cause mortality after simple and full adjustment. A sensitivity analysis excluding participants who died during the first year of follow-up revealed the following results for the highest quartile in the fully adjusted model: overall, HR (95%CI) = 1.54 (1.26–1.89); dysglycemia (prediabetes and T2DM), HR = 1.41 (1.15–1.73); T2DM, HR = 1.36 (1.10–1.67).

## Conclusion

We found CV-FPG to be useful for measurement of GV. It could also be used for the prognostic stratification of patients with dysglycemia.

## Introduction

Glycemic control among dysglycemic patients is usually assessed based on glycosylated hemoglobin (HbA1c), which reflects average blood glucose over previous months but does not inform about oscillations in blood sugar over time. Two patients with the same HbA1c may experience different glycemic excursions. For this reason, measurement of glycemic variability (GV) has been proposed as a tool for glucose monitoring in patients with type 1 diabetes (T1DM) and type 2 diabetes (T2DM) with severe insulin insufficiency [1] or associated conditions such as kidney failure. GV is defined as the oscillation of blood glucose levels outside the normal range. It can be classified into short-term variability (variations within the same day or between days) and long-term variability (oscillations between different clinical visits). The former is based on determinations obtained by continuous glucose monitoring and the latter on determinations of basal plasma glucose, HbA1c, or postprandial glucose obtained at different visits to the hospital or primary care center. The most widely used long-term GV measures are standard deviation and coefficient of variation, which are easy to calculate and interpret.

Many studies have evaluated whether fluctuations in glycemia are directly related to the incidence of diabetes complications. In their meta-analysis, Nalysnyk et al [2] found a link between GV, as measured by the coefficient of variation of fasting plasma glucose, and the risk of both microvascular and macrovascular complications, as well as mortality, among patients with T2DM. Specifically, an association with the development or progression of diabetic retinopathy, cardiovascular events, and mortality was evidenced in 9 of the 10 studies included. Similar findings were reported in a second meta-analysis evaluating variability in HbA1c [3], although most studies were retrospective and were poorly adjusted for potential confounders. Glycemic fluctuations have been reported to increase oxidative stress [4], inflammatory response, and endothelial damage, all of which would lead to vascular complications [5].

By contrast, the recent meta-analysis by Alatawi and Mirghani [6] included seven studies, of which four demonstrated an association between myocardial infarction and GV and three a neutral effect. Indeed, in their meta-analysis of studies on patients with DM, Smith-Palmer et al [7] showed that the association between GV and myocardial infarction was observed only among patients with T1DM.

Although GV is observed mainly in patients with diabetes, it also affects patients with prediabetes and normal blood glucose [8]. However, few studies have analyzed the relationship between GV and adverse health outcomes (cardiovascular events and mortality) among patients with differences in glycemic status, and even fewer have been performed in southern European countries with healthier lifestyles [9]. This lack of information could be due to the lower frequency of all-cause and cardiovascular mortality in southern Europe than in central

and northern Europe. However, the progressive aging of the population of southern Europe and the co-occurrence of dysglycemia at older ages require further studies in this line of research. We tested the hypothesis that GV predicts all-cause mortality regardless of glycemic status, after adjusting for established cardiovascular disease. If this hypothesis is confirmed, GV should be incorporated into the prognostic stratification of patients with dysglycemia.

## Methods

### Study population

We prospectively included 3,438 patients with T2DM from the first recruitment (year 2007) and 726 from the second recruitment (year 2011) of the MADIABETES cohort. These patients constitute the Spanish T2DM cohort with the highest number of person-years of follow-up. MADIABETES is also one of the few Spanish cohorts comprising primary care patients. Findings for the variables recorded—age, sex, time since diagnosis of diabetes, hypertension, dyslipidemia, and microvascular complications—are similar to those of Spanish hospital-based studies [10, 11] and the data reported by Bodicoat et al [12]. Likewise, 1,485 patients were included from the SPREDIA (Screening prediabetes and type 2 diabetes) cohort [13], which was initiated in 2010 and comprised 161 people with T2DM, 78 with previously unknown diabetes, 265 with prediabetes (impaired glucose tolerance [IGT]), and 981 with normal glycemia values. The main aim of the SPREDIA cohort study was to evaluate the performance of the Finnish Diabetes Risk Score (FINDRISC) and a simplified FINDRISC score (MADRISC) in screening for undiagnosed type 2 diabetes mellitus and dysglycemia. Given that 519 patients were excluded for having <3 fasting plasma glucose (FPG) measurements during follow-up, this analysis was based on 5,130 patients (Fig 1).

### Study variables

Patient follow-up started in 2007 and ended in 2019. Mortality data from the year 2020 were not included since values were substantially higher than expected because of COVID-19. As a dynamic cohort, not all patients completed the 12 years of follow-up, and the median follow-up for assessment of mortality was 9.8 years. Follow-up was terminated because of one of the following three circumstances: end of follow-up (12/31/2019), loss to follow-up through change of residence, and date of death. Mortality data were obtained from the Ministry of Health (National Institute of Deaths Registry [INDEF]), which includes the date of death but not its cause. There were no losses to follow-up with respect to mortality, because, regardless of whether the patient moved to a new city, mortality is recorded at national level and is based on the patient's identification data. These include the national identity card number, which is unique for each Spanish citizen.

In addition, demographic, anthropometric, clinical, and laboratory data were obtained from the family physician responsible for each patient and, when necessary, from the electronic primary care clinical records (AP-Madrid® software), which have been validated for research purposes [14] and are widely used in morbidity and mortality studies [15].

Participants not previously diagnosed with DM at baseline underwent a standard 75-g oral glucose tolerance test (OGTT), which was performed according to World Health Organization (WHO) recommendations [16]. The test includes FPG and glucose measurements over time; the glucose oxidase method was used to determine blood glucose levels. In patients who underwent an OGTT, the FPG was taken after eight hours of fasting and before glucose intake. General practitioners requested FPG measurements under conditions of usual clinical practice, and patients with less well-controlled disease tended to have a significantly higher number of annual blood glucose measurements. When this was the case, participating physicians

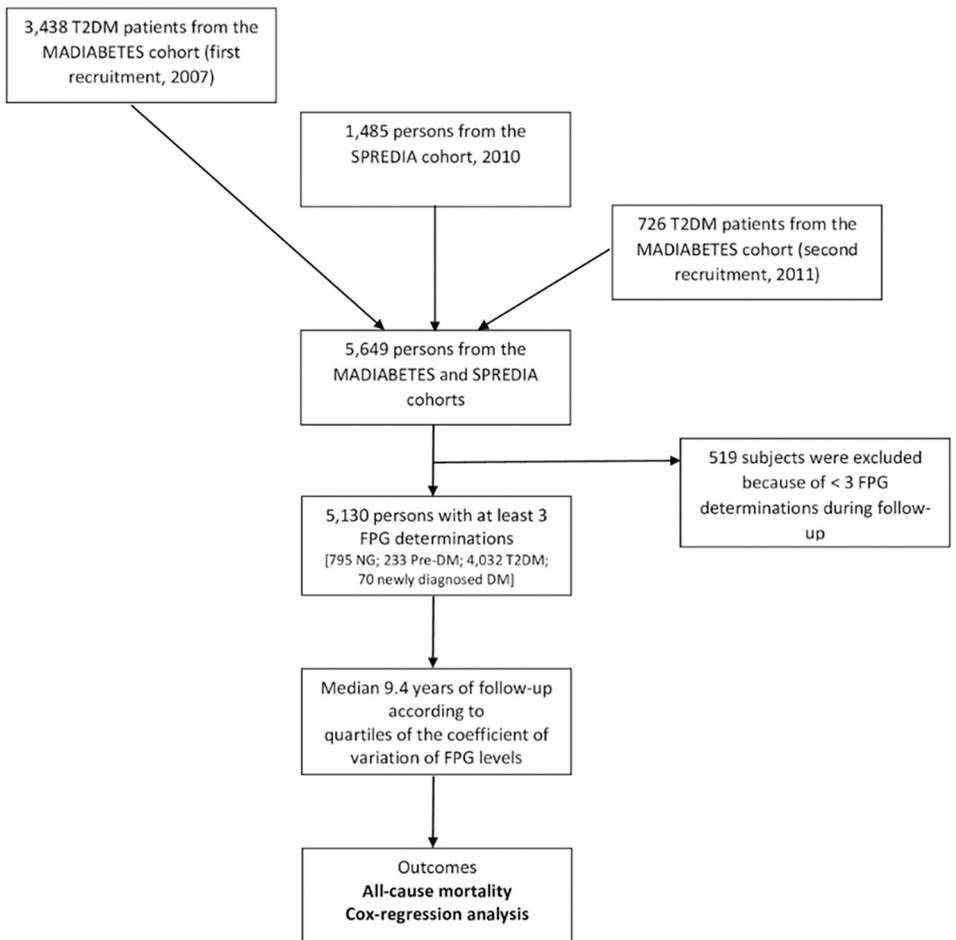

**Fig 1. Flow chart of study participants.** FPG: fasting plasma glucose; NG: normoglycemia; Pre-DM: prediabetes; T2DM: type 2 diabetes mellitus.

included only the last blood glucose measurement for the year in the data collection notebook. The median number of glucose measurements was eight. GV was only measured when at least three glucose measurements were collected.

Glycemic status was defined using the oral glucose tolerance test (OGTT) [17] as follows: normoglycemia (OGTT <140 mg/dl), prediabetes (OGTT 140–199 mg/dl), and newly diagnosed DM (participants with no previous diagnosis of DM at baseline and an OGTT ≥200 mg/dl).

Pre-existing cardiovascular disease was defined as a history of myocardial infarction, stroke, or peripheral vascular disease.

## Statistical analyses

As previously mentioned, the coefficient of variation of FPG (CV-FPG) was obtained in patients with at least three FPG values during follow-up (80.9%, 87.5%, and 93% in patients with normoglycemia, IGT, and T2DM, respectively) and calculated, for each patient, as the ratio of the standard deviation to the mean FPG multiplied by 100 (CV = SD/mean x 100, in %). A certain degree of GV is reasonable in subjects with normal glucose tolerance and even more so in those with diabetes or impaired glucose regulation in blood. Therefore, it is crucial

to identify the limit beyond which GV acquires pathological significance (association with mortality). For this reason, patients were categorized according to quartiles of CV-FPG, both in the overall sample and in the subsamples of patients with T2DM and patients with T2DM plus IGT, given that CV-FPG varies with each glycemic status category. The values of these quartiles for the total sample were as follows: Q1: ≤9.047; Q2: 9.048 to 15.232; Q3: 15.233 to 24.438: Q4: ≥24.439. For the subsample of patients from MADIABETES and SPREDIA (T2DM patients), the values were as follows: Q1: ≤12.3287; Q2: 12.3288 to 18.6246; Q3: 18.6247 to 27.0836; Q4: ≥27.0836.

Data are presented as proportions, means (standard deviation, SD), or, in the case of variables that did not conform to a normal distribution, medians (interquartile range). The $t$ test was used to compare two means, whilst the $\chi^2$ test was used for two or more proportions. An analysis of variance (ANOVA) was performed to compare continuous variables among the four quartiles of CV-FPG.

The mortality rate was calculated by considering the total number of deaths during follow-up divided by the total number of person-years.

Univariate survival analysis was performed using the Kaplan-Meier method and log-rank test. Multivariate survival analysis was conducted using Cox regression. In the first analysis with all subjects, the hazard ratios (HRs) and 95% confidence interval (CI) were calculated based on the following models: model 1, adjusted for age and sex; model 2, further adjusted for history of cardiovascular disease; model 3, further adjusted for glycemic status, hypertension, and use of statins, aspirin and antidiabetic-drugs in patients with T2DM; and model 4, further adjusted for baseline FPG in all samples and for baseline FPG plus average HbA1c (when at least two measurements were taken) in the T2DM sample. The interaction between CV-FPG and sex was assessed using a likelihood ratio test of their product terms in the full model for each glycemic status.

Lastly, a sensitivity analysis was performed excluding participants who died during the first year of follow-up to avoid the possible influence of the severity of underlying illnesses.

The analyses were performed with SPSS version 21.0 (IBM Corp., Armonk, NY, USA); a 2-sided p value < 0.05 was considered statistically significant.

## Ethics statement

The study was conducted according to the guidelines of the Declaration of Helsinki and approved by the Institutional Review Board of Ramón y Cajal Hospital (Madrid) for the MADIABETES cohort (approval identification code:2017/335) and the Institutional Review Board of Carlos III Hospital (Madrid) for the SPREDIA cohort (approval identification code: P07/2012). A written informed consent was obtained from all subjects involved in the study.

## Results

During follow-up, a total of 1,223 patients (657 men, 566 women) died, with an all-cause mortality rate of 23.35/1,000 person-years (26.07/1,000 in men and 20.83/1,000 in women). The mortality rates according to glycemic status are shown in Fig 2. Table 1 shows the baseline socio-demographic, anthropometric, and clinical findings for survivors and patients who died after a median of 9.8 years of follow-up. Compared with the survivors, patients who died were more likely to be male, older, ex-smokers, and hypertensive. They also had dyslipidemia and more frequently had a history of cardiovascular disease. They also had a lower mean body mass index, higher mean systolic blood pressure, and higher mean FPG.

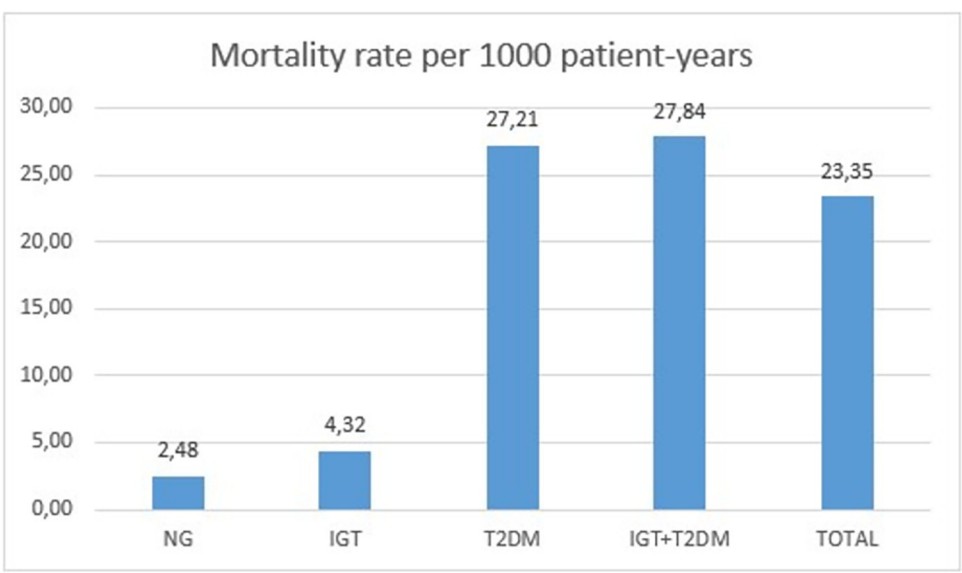

**Fig 2. Mortality rates according to glycemic status.** NG: normoglycemia; IGT: impaired glucose tolerance; T2DM: type 2 diabetes mellitus.

In Table 2 we summarize the baseline treatments received by study participants, finding significantly higher use of statins, aspirin and insulin among patients who had died than among survivors.

Table 3 presents the baseline characteristics of all study participants according to the quartiles of the CV-FPG. A significant linear trend (p<0.05) across the baseline quartiles was observed for the following variables: male sex, smoking status, dyslipidemia, hypertension, glycemic status, cardiovascular disease, and use of statins and aspirin.

**Table 1. Baseline socio-demographic, anthropometric, and clinical data of study participants overall and by survival status.**

| Variables | All patients (N = 5,130) | Dead (N = 1,223) | Survivors (N = 3,907) | p value |
|---|---|---|---|---|
| Age, mean (SD) | 66.6 (101) | 74.5 (8.7) | 64.1 (9.2) | <0.001 |
| Sex male, n (%) | 2,474 (48.2) | 657 (53.7) | 1,817 (46.5) | <0.001 |
| Never smoked, n (%) | 2,619 (51.1) | 694 (56.7) | 1,925 (49.3) | <0.001 |
| Ex-smoker, n (%) | 1,386 (27) | 346 (28.3) | 1,040 (26.6) | |
| Active smoker, n (%) | 1,125 (21.9) | 183 (15) | 942 (24.1) | |
| BMI, mean (SD) | 30.2 (5.1) | 29.7 (5.2) | 30.4 (5.1) | <0.001 |
| Baseline SBP, mean (SD) | 131.7 (11.3) | 133 (10.8) | 131.2 (11.4) | <0.001 |
| Baseline DBP, mean (SD) | 75.7 (6.9) | 73.4 (6.5) | 76.4 (6.8) | <0.001 |
| T2DM, n (%) | 4,102 (80.0) | 1,199 (98) | 2,903 (74.0) | <0.001 |
| Prediabetes (IGT), n (%) | 233 (4.5) | 8 (0.7) | 225 (5.8) | |
| Normoglycemia, n (%) | 795 (15.5) | 16 (1.3) | 779 (19.9) | |
| History of CVD, n (%) | 849 (16.5) | 400 (32.7) | 449 (11.5) | <0.001 |
| Hypertension, n (%) | 3,596 (70.1) | 1,015 (83) | 2,581 (66.1) | <0.001 |
| Dyslipidemia, n (%) | 2,669 (52) | 678 (55.4) | 1,991 (51) | 0.006 |
| Baseline FPG level, mean (SD) | 127.9 (31) | 134.8 (30.2) | 125.8 (31) | <0.001 |

BMI: body mass index; SBP: systolic blood pressure; DBP: diastolic blood pressure; T2DM: type 2 diabetes mellitus; IGT: impaired glucose tolerance; CVD: cardiovascular disease; FPG: fasting plasma glucose

**Table 2. Baseline treatments of study participants overall and by survival status.**

| Variables | All patients (N = 5,130) | Dead (N = 1,223) | Survivors (N = 3,907) | p value |
|---|---|---|---|---|
| Statin use, n (%) | 3,142 (61.2) | 819 (67) | 2,323 (59.5) | <0.001 |
| Aspirin use, n (%) | 2,611 (50.9) | 844 (69) | 1,767 (45.2) | <0.001 |
| Metformin use, n (%) | 2,128 (47.8) | 464 (48.2) | 1,664 (47.7) | 0.784 |
| Sulfonylurea use, n (%) | 864 (19.4) | 202 (21) | 662 (19) | 0.163 |
| Insulin use, n (%) | 982 (22.1) | 354 (36.8) | 628 (18) | <0.001 |
| GLP-1_receptor agonist use, n (%) | 31 (0.6) | 3 (0.2) | 28 (0.7) | 0.063 |
| SGLT-2 inhibitor use, n (%) | 48 (0.9) | 1 (0.1) | 47 (1.2) | <0.001 |
| DPP-4 inhibitor use, n (%) | 182 (3.5) | 51 (4.2) | 131 (3.4) | 0.178 |

GLP-1: glucagon like-peptide 1; SGLT-2: sodium-glucose co-transporter-2; DPP-4: dipeptidyl peptidase 4.

As shown in Fig 3, crude mortality was higher in the top quartile of CV-FPG than in the other quartiles (p< 0.001) during the 12-year follow-up period.

Table 4 shows all-cause mortality in total study participants according to quartiles of CV-FPG. Compared with patients in the lowest quartile, age- and sex-adjusted HRs (95% CI) in the third and highest CV-FPG quartiles were 1.31 (1.07–1.60) and 1.87 (1.54–2.26), respectively. However, findings for the second quartile were not significant (HR [95%CI]

**Table 3. Baseline factors of 5,130 subjects with differences in glycemic status grouped by CV-FPG quartile.**

| Variables | CV-FPG Quartile | | | | |
|---|---|---|---|---|---|
| | 1 (lowest) | 2 | 3 | 4 (highest) | p value |
| Range | ≤9.047 | 9.048–15.232 | 15.233–24.438 | ≥24.439 | |
| N | 1,282 | 1,283 | 1,283 | 1,282 | |
| **Anthropometric and clinical variables** | | | | | |
| Male sex, n (%) | 570 (44.5) | 628 (48.9) | 635 (49.5) | 641 (50) | 0.01* |
| Age, mean (SD) | 64.7 (8.3) | 66.8 (9.6) | 67.7 (10.2) | 67.2 (11.7) | <0.01 |
| BMI, mean (SD) | 29.0 (4.9) | 30.4 (4.9) | 30.5 (5.0) | 30.7 (5.4) | <0.01 |
| Baseline SBP, mean (SD) | 129.2 (12.2) | 131.8 (10.8) | 132.3 (10.5) | 133.1 (11.2) | <0.01 |
| Baseline DBP, mean (SD) | 76.1 (7.4) | 76.0 (6.7) | 75.4 (6.6) | 75.3 (6.8) | 0.01 |
| Smoking, n (%) | 364 (28.4) | 273 (21.3) | 231 (18.0) | 257 (20.0) | <0.01* |
| Hypertension, n (%) | 680 (53) | 934 (72.8) | 995 (77.6) | 987 (77) | <0.01* |
| Dyslipidemia, n (%) | 633 (49.4) | 675 (52.6) | 650 (50.7) | 711 (55.5) | <0.01* |
| **Glycemic status** | | | | | |
| Normoglycemia, n (%) | 609 (47.5) | 156 (12.2) | 21 (1.6) | 9 (0.7) | <0.01* |
| Prediabetes (IGT), n (%) | 148 (11.5) | 65 (5.1) | 14 (1.1) | 6 (0.5) | |
| Type 2 DM, n (%) | 525 (41.0) | 1,062 (82.8) | 1,248 (97.3) | 1,267 (98.8) | |
| **Cardiovascular Disease** | | | | | |
| Previous myocardial infarction, n (%) | 59 (4.6) | 108 (8.4) | 153 (11.9) | 161 (12.6) | <0.01* |
| Previous stroke, n (%) | 33 (2.6) | 72 (5.6) | 98 (7.6) | 107 (8.3) | <0.01* |
| Primary prevention, n (%) | 1,181 (92.1) | 1.091 (85.0) | 1,028 (80.1) | 981 (76.5) | <0.01* |
| **Medication profile** | | | | | |
| Statin use, n (%) | 574 (44.8) | 798 (62.2) | 873 (68.0) | 897 (70.0) | <0.01* |
| Aspirin use, n (%) | 335 (26.1) | 626 (48.8) | 794 (61.9) | 856 (66.8) | <0.01* |

*p value for linear trend across baseline CV-FPG quartiles.

CV-FPG: coefficient of variation of fasting plasma glucose; BMI: body mass index; SBP: systolic blood pressure; DBP: diastolic blood pressure; T2DM: type 2 diabetes mellitus; IGT: impaired glucose tolerance

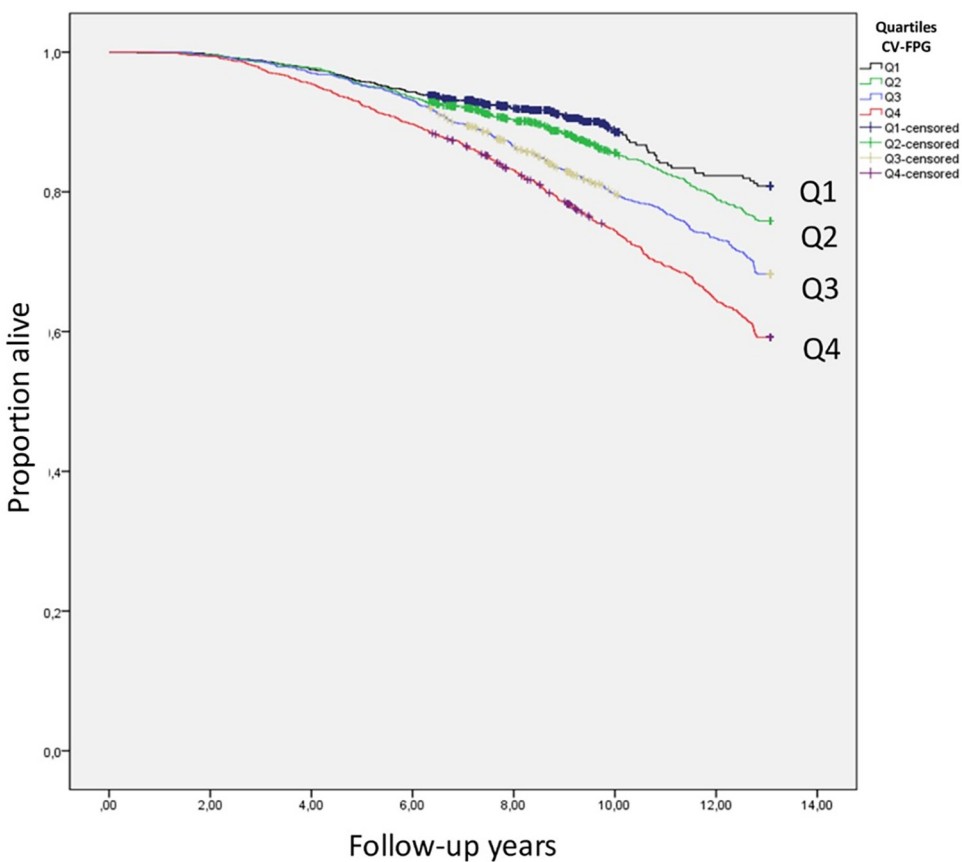

**Fig 3. Crude all-cause mortality according to quartiles of the coefficient of variation of fasting plasma glucose levels (5,130 subjects with normoglycemia, prediabetes, or T2DM).**

**Table 4. Hazard ratios of all-cause mortality grouped by CV-FPG in patients with normoglycemia, IGT and T2DM.**

| Normoglycemia, IGT and T2DM (n = 5,130) | CV-FPG quartile | | | |
|---|---|---|---|---|
| | 1 (lowest) | 2 | 3 | 4 (highest) |
| N | 1,282 | 1,283 | 1,283 | 1,282 |
| All-cause mortality, n (%) | 137 (11.2) | 238 (19.5) | 365 (29.8) | 483 (39.5) |
| Person-years | 11,370 | 13,270 | 14,020 | 13,710 |
| Mortality rate (per 1,000 person-years) | 12.05 | 17.94 | 26.03 | 35.23 |
| **Model 1** | 1 | 1.03 (0.83–1.27) | 1.31 (1.07–1.60)* | 1.87 (1.54–2.26)** |
| **Model 2** | 1 | 0.98 (0.79–1.21) | 1.23 (1.01–1.50)* | 1.71 (1.41–2.08)** |
| **Model 3** | 1 | 0.84 (0.68–1.05) | 1.04 (0.85–1.27) | 1.44 (1.18–1.76)** |
| **Model 4** | 1 | 0.85 (0.70–1.05) | 1.01 (0.82–1.24) | 1.13–1.71)* |

Model 1: adjusted for age and sex. Model 2: adjusted for age, sex, and history of cardiovascular disease. Model 3: adjusted for variables in model 2 plus glycemic status, hypertension, use of aspirin and statins. Model 4: Model 3 plus baseline FPG

*p < 0.05

**p < 0.001.

CV-FPG: Coefficient of variation of fasting plasm glucose

**Table 5. Hazard ratios of all-cause mortality grouped by CV-FPG in T2DM patients.**

| T2DM (n = 4,102) | CV-FPG quartile | | | |
|---|---|---|---|---|
| | 1 (lowest) | 2 | 3 | 4 (highest) |
| N | 1,025 | 1,026 | 1,026 | 1,025 |
| All-cause mortality, n (%) | 227 (22.1) | 249 (24.3) | 310 (30.2) | 413 (40.3) |
| Person-years | 10,573 | 11,345 | 11,253 | 10,887 |
| Mortality rate (per 1,000 person-years) | 21.47 | 21.95 | 27.55 | 37.94 |
| **Model 1** | 1 | 0.91 (0.76–1.09) | 1.19 (1.01–1.41)* | 1.52 (1.25–1.86)** |
| **Model 2** | 1 | 0.82 (0.66–1.02) | 1.02 (0.83–1.25) | 1.42 (1.16–1.73)** |
| **Model 3** | 1 | 0.82 (0.66–1.02) | 1.02 (0.83–1.25) | 1.41 (1.16–1.73)** |
| **Model 4** | 1 | 0.82 (0.66–1.02) | 0.99 (0.80–1.22) | 1.37 (1.11–1.68)* |

Model 1: adjusted for age and sex. Model 2: adjusted for age, sex, and history of cardiovascular disease. Model 3: adjusted for variables in model 2 plus hypertension, use of aspirin and statins. Model 4: Model 3 plus use of GLP-1 receptor agonists, SGLT-2 inhibitors and DPP-4 inhibitors, baseline FPG and average HbA1c (when at least two measurements) *p < 0.05

**p < 0.001.

CV-FPG: Coefficient of variation of fasting plasm glucose; GLP-1: glucagon like-peptide-1; SGLT-2:sodium-glucose co-transporter-2; DPP-4: dipeptidyl peptidase 4

= 1.03 [0.83–1.27]). After further adjustment for history of cardiovascular disease (model 2), the effect of CV-FPG was attenuated, although it remained statistically significant for the third and fourth quartiles (HR [95%CI] = 1.23 [1.01–1.50] and 1.71 [1.41.2.08], respectively). Further adjustment (model 3 and 4) revealed a significant effect only for the highest quartile.

Table 5 shows the results of the main analysis only in patients with T2DM; the results are very similar, with significant HRs for the fourth quartile in all models and the third quartile in model 1. No significant interaction with sex was found (p = 0.957).

Lastly, Table 6 (prediabetes [IGT] or T2DM) also shows the significant effect of the fourth quartile on all-cause mortality in all models.

The results of the sensitivity analysis were consistent with previous findings. Therefore, the results for the highest quartile in the most adjusted model were as follows: overall, HR (95% CI) = 1.54 (1.26–1.89); dysglycemia (prediabetes and T2DM), HR = 1.41 (1.15–1.73); and T2DM, HR = 1.36 (1.10–1.67).

**Table 6. Hazard ratios of all-cause mortality grouped by CV-FPG in patients with IGT and T2DM.**

| IGT and T2DM (n = 4,335) | CV-FPG quartile | | | |
|---|---|---|---|---|
| | 1 (lowest) | 2 | 3 | 4 (highest) |
| N | 1,084 | 1,084 | 1,084 | 1,083 |
| All-cause mortality, n (%) | 205 (18.9) | 246 (22.7) | 325 (30.0) | 431 (39.8) |
| Person-years | 10,684 | 11,825 | 11,889 | 11,503 |
| Mortality rate (per 1,000 person-years) | 19.19 | 20.80 | 27.34 | 37.47 |
| **Model 1** | 1 | 0.89 (0.72–1.11) | 1.13 (0.92–1.39) | 1.61 (1.32–1.96)* |
| **Model 2** | 1 | 0.86 (0.69–1.07) | 1.08 (0.88–1.32) | 1.49 (1.22–1.82)* |
| **Model 3** | 1 | 0.86 (0.69–1.06) | 1.07 (0.87–1.31) | 1.48 (1.22–1.81)* |
| **Model 4** | 1 | 0.85 (0.68–1.06) | 1.03 (0.83–1.27) | 1.41 (1.15–1.73)* |

Model 1: adjusted for age and sex. Model 2: adjusted for age, sex, and history of cardiovascular disease. Model 3: adjusted for variables in model 2 plus hypertension, use of aspirin and statins. Model 4: Model 3 plus baseline FPG *p < 0.001.

CV-FPG: Coefficient of variation of fasting plasm glucose

## Discussion

To our knowledge, this is the first large cohort study performed in southern Europe to investigate the association between GV and all-cause mortality in patients with differences in glycemic status. This is the main difference with respect to previous, similar studies [18], given that we studied the GV in different metabolic situations (normoglycemia, impaired glucose tolerance [IGT], and diabetes mellitus).

Our findings show that the highest degree of GV, expressed as the highest quartile of the CV-FPG, behaves as a long-term predictor of all-cause mortality in patients with T2DM and the subgroup with prediabetes or T2DM for any adjusted model. In addition, the third quartile proved to be a predictor of mortality exclusively for the age- and sex-adjusted model in patients with T2DM.

Although there are different ways of measuring GV, the CV-FPG and mean amplitude of glycemic excursions (MAGE) are considered the most useful for research purposes [19]. Other studies have found similar results to ours with the CV-FGP. For example, the Verona Diabetes Study [20] enrolled 1,409 T2DM patients aged 56–74 years with a 10-year follow-up of mortality. In the multivariate analysis, the relative risk for all-cause death associated with the highest versus lowest tertile of the CV-FPG was 1.68 (95%CI, 1.29–2.18). The crude Kaplan-Meier analysis showed that survival was longer in patients in the lower tertile of CV-FPG (p = 0.001) than in patients of the other two tertiles, for whom differences were not significant. A subsequent analysis in 1,319 T2DM patients of the Verona Diabetes Study [21] showed differences between the age groups. The group aged >65 years had an adjusted HR for CV-FPG of 1.56 (1.17–2.08), and the younger group had a non-significantly adjusted HR of 1.34 (0.79–2.27). These differences are not surprising, given that the lower incidence of mortality and the smaller size in the younger group might have diminished the statistical power and thus precluded identification of significant results. Therefore, in our study, we preferred to adjust for age and not to stratify.

In a retrospective cohort study of 5,008 T2DM patients from Taiwan [22], the fully adjusted HR of all-cause mortality for the highest versus lowest tertile of annual CV-FPG was 5.53 (95% CI, 3.85–7.94). The association was considerably stronger than in our study, probably because participants were exclusively patients with T2DM treated in hospital, thus leading to selection bias.

In critically ill patients receiving intravenous insulin the coefficient of GV was independently associated with 30-day mortality (OR = 1.23 for every 10% increase, p<0.001), even after adjustment for hypoglycemia, age, disease severity, and comorbidities. The association was observed both in non-diabetics (OR = 1.37, p<0.001) and in diabetics (OR = 1.15, p = 0.001) [23]. The effect on mortality has been attributed to hypoglycemia, especially in critically ill patients, although our study ruled out this possibility, as it was adjusted for the presence of hypoglycemia (defined as <60 mg/dL). However, in chronic T2DM patients, the effect of GV on all-cause mortality is due to its association with 8-iso prostaglandin F2$\alpha$, a marker of oxidative stress and a potential mediator of organ dysfunction [4].

The association between mortality and GV could be due to an increase in the incidence of cancer, given the known association between marked GV and a dose-dependent high risk of future malignancies among people without diabetes [24]. Furthermore, both dysglycemia and overt atherosclerosis increase the risk of cancer [25].

A retrospective Chinese study [26] of 8,871 patients with T2DM followed for 7 years showed no association between CV-FPG and all-cause mortality after adjustment for baseline FPG. Furthermore, after stratifying by HbA1c, the HR of the highest CV-FPG quartile was only significant among those with HbA1c >7% (HR = 1.63; 95%CI, 1.25–2.13). In contrast,

our study included baseline FPG in the fully adjusted models; therefore, the significant results were independent of the degree of glycemic control.

Study results can be influenced by the method used to measure GV. In this regard, the ADVANCE trial [27] in T2DM patients analyzed the visit-to-visit GV using the SD of HbA1c (SD-HbA1c) and of glucose (SD-FPG). There were significant linear associations between SD-HbA1c and combined macro-and microvascular events, major macrovascular events, and all-cause mortality after adjusting for mean HbA1c during the first 24 months and other confounders. SD-FPG, adjusted for mean FPG during the first 24 months and other factors, was also continuously associated with combined macro- and microvascular events, major macrovascular events, and major microvascular events, but not for all-cause mortality. In our case, we could not compare the results with variability in HbA1c, because this measurement was recorded in very few patients with normoglycemia.

In contrast, in an observational analysis of the ALLHAT study [28] (4,982 hypertensive participants, 35.3% diagnosed with DM and 25.6% with a history of cardiovascular disease), the fully-adjusted HR (95% CI) for all-cause mortality was 2.22 (1.22–4.04) for the highest versus lowest quartile of SD-FPG ($\geq$26.4 vs. <5.5 mg/dL). A Taiwanese retrospective cohort study in T2DM patients [29] followed for at least 2 years showed that variability in HbA1c, as measured using SD-HbA1c or CV-HbA1c, was a significant risk factor for all-cause mortality, yielding a higher HR with SD-HbA1c than with CV-HbA1c (1.99 vs. 1.06, both with p<0.05) after full adjustment for use of statins, as in the present study.

Our study is characterized by a series of strengths, including its prospective design and the large number of patients with diabetes, prediabetes, and normoglycemia, as well as its long-term follow-up. In addition, to our knowledge, ours is the first study to examine the relationship between variability in FPG and all-cause mortality in patients with differences in glycemic status in southern European countries. This aspect is especially relevant, given the possible lower effect of GV on all-cause mortality in countries with healthier lifestyles [9] and better glycemic control than other countries participating in the EUROASPIRE IV survey [30].

However, our study is also subject to a series of limitations. First, given that we included patients with differences in glycemic status, the analyses could not be adjusted for duration of diabetes, mean HbA1c, diabetic nephropathy, diabetes treatments, or microalbuminuria, as in other studies. Second, we did not have information on the cause of death, which would have enabled us to verify that mortality is, to a large extent, accounted for by cardiovascular disease, given the known association between GV and macrovascular complications. Third, we did not record hypoglycemia episodes and were therefore unable to assess their association with mortality. Fourth, we could not study GV measured with CV-HbA1c, given that few persons with normoglycemia or IGT had at least three HbA1c measurements during follow-up. Fifth, as data were from two sources, namely, the MADIABETES and SPREDIA cohorts, they may have been subject to a certain degree of heterogeneity.

Lastly, given the observational nature of the present study, individuals with higher GV and lower GV were dissimilar. Therefore, to obtain an accurate picture of the association between GV and all-cause mortality, it was necessary to adjust for differences in both groups in the multivariate analysis. Propensity score matching (PSM) would be a more appropriate alternative that would yield less biased results than standard methods such as Cox regression. However, one of the drawbacks of PSM is the loss of sample in terms of size. In addition, PSM should not be used in practice because our sample size was insufficiently large. Given that propensity scores can only control for observed confounders, they cannot be counted upon to balance unobserved covariates.

## Conclusion

Our results and those of other, similar studies show that the prognostic stratification of patients with some degree of dysglycemia should incorporate measurement of GV. CV-FPG proved useful for measuring GV in our study. However, the best method for assessing GV under conditions of daily clinical practice remains to be defined. In addition, it remains unclear whether the consequences of GV for mortality can justify using drugs for control of GV, especially in prediabetic patients.

## Acknowledgments

This study was possible thanks to the physicians and nurses of the MADIABETES and SPRE-DIA Research Groups: V Iriarte-Campo (Foundation for Research and Biomedical Innovation of Primary Care of the Community of Madrid (FIIBAP)), C de Burgos-Lunar (Clínico San Carlos University Hospital), AM Sobrado-de Vicente-Tutor (Alameda de Osuna Health Center), M Sanz-Pascual (Alameda de Osuna Health Center), D Beamud-Victoria (Felipe II Health Center, Móstoles), M Arnalte-Barrera (Andrés Mellado Health Center), S Pulido-Fernández (North Multiprofessional Teaching Unit, Madrid), EM Donaire-Jiménez (Artilleros Health Center), C Montero-Lizana (Ciudad Jardín Health Center), M Domínguez-Paniagua (Barajas Health Center), P Serrano-Simarro (Barajas Health Center), R Echegoyen-de Nicolás (Nuñez Morgado Health Center), P Gil-Díaz (Benita de Ávila Health Center), I Cerrada-Somolinos (Campamento Health Center), R Martín-Cano (Canillejas Health Center), A Cava-Rosado (Ciudad Jardín Health Center), T Mesonero-Grandes (Ciudad Jardín Health Center), E Gómez-Navarro (Estrecho de Corea Health Center), A Maestro-Martín (Fronteras Health Center, Torrejón), A Muñoz-Cildoz (Fronteras Health Center, Torrejón), ME Calonge-García (Barcelona Health Center), M Martín-Bun (General Fanjul Health Center), P Carreño-Freire (General Fanjul Health Center), J Fernández-García (Peña Prieta Health Center), A Morán-Escudero (Juncal Health Center, Torrejón), J Martínez-Irazusta (Brújula Health Center, Torrejón), E Calvo-García (Cerceda Health Center, El Boalo), AM Alayeto-Sánchez (Mar Báltico Health Center), C Reyes-Madridejos (Rosa de Luxemburgo Health Center, San Sebastián de los Reyes), MJ Bedoya-Frutos (María Jesús Hereza Health Center, Leganés), B López-Sabater (Meco Health Center, Meco), J Innerarity-Martínez (María Jesús Hereza, Leganés), A Rosillo-González (Sector Embarcaciones Health Center, Tres Cantos), AI Menéndez-Fernández (Ensanche de Vallecas Health Center), F Mata-Benjumea (Potosí Health Center), C Martín-Madrazo (Monóvar Health Center), MJ Gomara-Martínez (Cortes Health Center), C Bello-González (La Chopera Health Center, Alcobendas), A Pinilla-Carrasco (Reyes Magos Health Center, Alcalá de Henares), M Camarero-Shelly (Barajas Health Center), A Cano-Espin, (García Noblejas Health Center) J Castro-Martin (Benita de Ávila Health Center), B de Llama-Arauz (Entrevías Health Center), A de Miguel-Ballano (Reyes Magos Health Center, Alcalá de Henares), MA García-Alonso (Foundation for Research and Biomedical Innovation of Primary Care of the Community of Madrid (FIIBAP)), JN García-Pascual (Arganda-Felicidad Health Center, Arganda del Rey), MI González-García (Barajas Health Center), C López-Rodríguez (Ciudad Jardín Health Center), M Miguel-Garzón (Juan de Austria Health Center, Alcalá de Henares), MC Montero-García (Barajas Health Center), JC Abanades-Herranz (Monóvar Health Center), S Muñoz-Quiros-Aliaga (Monóvar Health Center), S Núñez-Palomo (Torrelaguna Health Center, Torrelaguna), O Olmos-Carrasco (Jazmín Health Center), N Pertierra-Galindo (Monóvar Health Center), G Reviriego-Jaén (Barajas Health Center), P Rius-Fortea (Valdezarza Health Center), G Rodríguez-Castro (Foundation for Research and Biomedical Innovation of Primary Care of the Community of Madrid (FIIBAP)), JM San Vicente-Rodríguez (Barajas Health Center), ME Serrano-Serrano (Los Fresnos Health Center,

Torrejón), MM Zamora-Gómez (Daroca Health Center), MP Zazo-Lázaro (Torito Health Center), L Pérez-Isla (Clínico San Carlos University Hospital), I Vicente (Monóvar Health Center), S Artola (José Marvá Health Center), MI Granados-Menéndez (Monóvar Health Center), I Dujovne-Kohan (Foundation for Research and Biomedical Innovation of Primary Care of the Community of Madrid (FIIBAP)), RM Chico-Moraleja (Gómez Ulla, Defense Central Hospital), C Martín-Madrazo (Monóvar Health Center), C Aguilera-Linde (Ciudad de los Periodistas Health Center), AR Aguirre-De Carcer-Escolano (Ventilla Health Center), P Alonso-Sacristán (Ciudad de los Periodistas Health Center), MJ Álvarez-Otero (Dr. Castroviejo Health Center), P Arribas-Pérez (Santa Hortensia Health Center), ML Asensio-Ruiz (Fuentelarreina Health Center), P Astorga-Díaz (Barrio del Pilar Health Center), B Berriatua-Ena (Dr. Castroviejo Health Center), AI Bezos-Varela (José Marvá Health Center), MJ Calatrava-Triguero (Ciudad Jardín Health Center), C Casanova-García (Barrio del Pilar Health Center), A Conde-Llorente (Barrio del Pilar Health Center), C Díaz-Laso (Fuentelarreina Health Center), E Elviro-García (Ciudad de los Periodistas Health Center), O Enríquez-Dueñas (Fuentelarreina Health Center), MI Ferrer-Zapata (El Greco Health Center, Getafe), F Antuña (María de Guzmán Health Center, Alcalá de Henares), MI García-Lázaro (Ciudad de los Periodistas Health Center), MT Gómez-Rodríguez (Cerro del Aire Health Center, Majadahonda), A Gómez-Lucena (La Ventilla Health Center), F Herrero-Hernández (Dr. Castroviejo Health Center), R Julián-Viñals (Dr. Castroviejo Health Center), G López-Ruiz Ogarrio (in memoriam), MC Lumbreras-Manzano (Ibiza Health Center), SP Luquero-López (Ciudad de los Periodistas Health Center), A Martínez-Cabrera-Peláez (Barrio del Pilar Health Center), M Nieto-Candenas (La Ventilla Health Center), MA Rabanal-Carrera (Barrio del Pilar Health Center), A Castellanos-Rodríguez (Ciudad de los Periodistas Health Center), A López-Castellanos (Orcasur Health Center), M Velázquez-García (Barrio del Pilar Health Center), and M Ruiz-Pacheco (Algete Health Center, Algete).

## Author Contributions

**Conceptualization:** Miguel A. Salinero-Fort.

**Data curation:** Miguel A. Salinero-Fort, Juan Cárdenas-Valladolid.

**Formal analysis:** Miguel A. Salinero-Fort, F. Javier San Andrés-Rebollo.

**Funding acquisition:** Miguel A. Salinero-Fort, F. Javier San Andrés-Rebollo.

**Methodology:** Miguel A. Salinero-Fort, F. Javier San Andrés-Rebollo, Juan Cárdenas-Valladolid.

**Resources:** José M. Mostaza, Carlos Lahoz, Fernando Rodriguez-Artalejo, Paloma Gómez-Campelo, Pilar Vich-Pérez, Rodrigo Jiménez-García, Ana López de Andrés, José M. de Miguel-Yanes.

**Supervision:** Miguel A. Salinero-Fort, José M. Mostaza, Carlos Lahoz, Fernando Rodriguez-Artalejo, Paloma Gómez-Campelo, Pilar Vich-Pérez, Rodrigo Jiménez-García, Ana López de Andrés, José M. de Miguel-Yanes.

**Validation:** Juan Cárdenas-Valladolid.

**Writing – original draft:** Miguel A. Salinero-Fort, José M. Mostaza, Carlos Lahoz, Fernando Rodriguez-Artalejo, Paloma Gómez-Campelo, Pilar Vich-Pérez, Rodrigo Jiménez-García, Ana López de Andrés, José M. de Miguel-Yanes.

**Writing – review & editing:** Miguel A. Salinero-Fort, F. Javier San Andrés-Rebollo, Juan Cárdenas-Valladolid, José M. Mostaza, Carlos Lahoz, Fernando Rodriguez-Artalejo, Paloma

Gómez-Campelo, Pilar Vich-Pérez, Rodrigo Jiménez-García, Ana López de Andrés, José M. de Miguel-Yanes.

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
