## [Decision Letter · Decision Letter 0]

20 Apr 2022

PONE-D-21-35463Glycemic variability and all-cause mortality in a large prospective southern European cohort of patients with differences in glycemic status.PLOS ONE

Dear Dr. Salinero-Fort,

Thank you for submitting your manuscript to PLOS ONE. After careful consideration, we feel that it has merit but does not fully meet PLOS ONE’s publication criteria as it currently stands. Therefore, we invite you to submit a revised version of the manuscript that addresses the points raised during the review process.

ACADEMIC EDITOR: Please address the questions raised by the reviewer-1Please ensure that your decision is justified on PLOS ONE’s publication criteria and not, for example, on novelty or perceived impact.

We look forward to receiving your revised manuscript.

Kind regards,

Venkata Naga Srikanth Garikipati, PhD

Academic Editor

PLOS ONE

2. Please provide additional details regarding participant consent. In the Methods section,

please ensure that you have specified (1) whether consent was informed and (2) what type you obtained

(for instance, written or verbal). If your study included minors, state whether you obtained consent from parents

or guardians. If the need for consent was waived by the ethics committee, please include this information.

“This study was funded by Instituto de Salud Carlos III through projects “PI15/00259” and “PI18/01025” and co-funded by the European Regional Development Fund, “A way of shaping Europe””

“The authors declare that they have no known competing financial interests or personal relationships that could influence the work reported in this paper.”

6. We note that you have indicated that data from this study are available upon request. PLOS only allows data to be available upon request if there are legal or ethical restrictions on sharing data publicly. For more information on unacceptable data access restrictions, please see http://journals.plos.org/plosone/s/data-availability#loc-unacceptable-data-access-restrictions.

7. Please note that in order to use the direct billing option the corresponding author must be affiliated with the chosen institute. Please either amend your manuscript to change the affiliation or corresponding author, or email us at plosone@plos.org with a request to remove this option.

8. One of the noted authors is a group or consortium [Abanades-Herranz JC]. In addition to naming the author group, please list the individual authors and affiliations within this group in the acknowledgments section of your manuscript. Please also indicate clearly a lead author for this group along with a contact email address.

Reviewers' comments:

Reviewer's Responses to Questions

**Comments to the Author**

1. Is the manuscript technically sound, and do the data support the conclusions?

Reviewer #1: Yes

Reviewer #2: Yes

2. Has the statistical analysis been performed appropriately and rigorously? 

Reviewer #1: No

Reviewer #2: Yes

3. Have the authors made all data underlying the findings in their manuscript fully available?

Reviewer #1: Yes

Reviewer #2: Yes

4. Is the manuscript presented in an intelligible fashion and written in standard English?

Reviewer #1: No

Reviewer #2: Yes

5. Review Comments to the Author

Reviewer #1: The authors have found in their study that CV-FPG is useful for measurement of GV. Although the finding are potential, reviewer has few concerns:

-data representation should be re-arranged. For instance, few table are too long to read. Split them into two.

- Consult the statistician to confirm the statistical tests used in the study.

- Check for the grammar and typo error.

-Please report the other medical underlyig conditions and how they affect the finding?

Reviewer #2: The current study “Glycemic variability and all-cause mortality in a large prospective southern European cohort of patients with differences in glycemic status” contributes in highlighting the significance to include glycemic variability (GV) as one of the factors to monitor the prognosis of different glycemic status patients.

The finding do suggest how GV could behave as a long-term predictor of all-cause mortality in individuals with certain underlying conditions with variation included .It is rightly understood, with the limitation of the method to satisfy all the variables not limited to duration of the disease, diabetes treatments, or microalbuminuria etc, which would prevent this to be used with confidence.

But the authors do make a valid case for inclusion of GV analysis through their proposed methods so as to derive information for managing the outcome of the diabetic condition of the patients.

There are however, some concerns with the current draft of the manuscript. The authors should follow a consistent mode of representation. E.g. at some places Fig. is used while at some places Figure is written. Kindly adhere to one form of writing.

6. PLOS authors have the option to publish the peer review history of their article (what does this mean?). If published, this will include your full peer review and any attached files.

Reviewer #1: No

Reviewer #2: No

---

## [Author Response · Author response to Decision Letter 0]

2 Jun 2022

We thank the editor and the two reviewers for their comments on our manuscript entitled “Glycemic variability and all-cause mortality in a large prospective southern European cohort of patients with differences in glycemic status “. Below is our response to each point raised by the academic editor and reviewers. 

We hope that we satisfyingly addressed them, and that the manuscript will be now suited for publication. 

Sincerely, 

Salinero-Fort MA

On behalf of all authors,

Academic editor:

File naming was edited to comply with the style requirements. We hopefully have no divergences from the style requirements now.

2. Please provide additional details regarding participant consent. In the Methods section, please ensure that you have specified (1) whether consent was informed and (2) what type you obtained (for instance, written or verbal). If your study included minors, state whether you obtained consent from parents or guardians. If the need for consent was waived by the ethics committee, please include this information.

In methods section include the following paragraph:

“The study was conducted according to the guidelines of the Declaration of Helsinki and approved by the Institutional Review Board of Ramón y Cajal Hospital (Madrid) for the MADIABETES cohort (approval identification code:2017/335) and the Institutional Review Board of Carlos III Hospital (Madrid) for the SPREDIA cohort (approval identification code: P07/2012). A written informed consent was obtained from all subjects involved in the study”.

The e-mail of Institutional Review Board of Ramón y Cajal Hospital (Madrid) is: ceic.hrc@salud.madrid.org

“This study was funded by Instituto de Salud Carlos III through projects “PI15/00259” and “PI18/01025” and co-funded by the European Regional Development Fund, “A way of shaping Europe””

We confirm that:

“The authors declare that they have no known competing financial interests or personal relationships that could influence the work reported in this paper.”

We confirm that:

The authors have declared that no competing interests exist

There are restrictions on the availability of data for the MADIABETES study, due to the signed consent agreements around data sharing, which only allow access to external researchers for research following the project purposes. 

Requestors wishing to access the MADIABETES data used in this study can request it to the MADIABETES Steering Committee: estudios.fiibap@salud.madrid.org

The request will then be passed to members of the MADIABETES Steering Committee for deliberation.

However, the minimal data set is available from the Institutional Repository of the Regional Health System of the Community of Madrid (https://repositoriosaludmadrid.es/)

https://repositoriosaludmadrid.es/handle/20.500.12530/326/browse?type=author&order=ASC&rpp=20&authority=e55f8892-3241-4a96-a0d6-640436d5c75e&value=SALINERO-FORT%2C+MIGUEL+A.

6. We note that you have indicated that data from this study are available upon request. PLOS only allows data to be available upon request if there are legal or ethical restrictions on sharing data publicly. For more information on unacceptable data access restrictions, please see http://journals.plos.org/plosone/s/data-availability#loc-unacceptable-data-access-restrictions.

There are restrictions on the availability of data for the MADIABETES study, due to the signed consent agreements around data sharing, which only allow access to external researchers for research following the project purposes. 

Requestors wishing to access the MADIABETES data used in this study can request it to the MADIABETES Steering Committee: estudios.fiibap@salud.madrid.org

The request will then be passed to members of the MADIABETES Steering Committee for deliberation.

7. Please note that in order to use the direct billing option the corresponding author must be affiliated with the chosen institute. Please either amend your manuscript to change the affiliation or corresponding author, or email us at plosone@plos.org with a request to remove this option.

Corresponding author:

Salinero-Fort MA1-5

1. Foundation for Research and Biomedical Innovation of Primary Care of the Community of Madrid (FIIBAP) [payment institution]

2. The Hospital La Paz Institute for Health Research (IdiPAZ). 

3. Health Services and Chronic Conditions Research Network (REDISSEC), Madrid, Spain.

4. General Subdirectorate of Research and Documentation. Department of Health, Madrid, Spain.

5. Alfonso X El Sabio University, Madrid, Spain.

8. One of the noted authors is a group or consortium [Abanades-Herranz JC]. In addition to naming the author group, please list the individual authors and affiliations within this group in the acknowledgments section of your manuscript. Please also indicate clearly a lead author for this group along with a contact email address.

The MADIABETES consortium is currently led by Dr. Iriarte-Campo V, a researcher at the Foundation for Research and Biomedical Innovation of Primary Care of the Community of Madrid (FIIBAP). His e-mail is: victor.iriarte@salud.madrid.org

Dr. Abanades-Herranz since 2020 is not the leader or head of the group.

Reviewer #1: The authors have found in their study that CV-FPG is useful for measurement of GV. Although the finding are potential, reviewer has few concerns:

-data representation should be re-arranged. For instance, few table are too long to read. Split them into two.

As suggested by the reviewer, the previous table 1 is now broken down into two new tables and the previous table 3 is now broken down into three new tables. This will surely make the tables easier to read.

- Consult the statistician to confirm the statistical tests used in the study.

The normality of the distribution of all the continuous variables was checked using the Kolmogorov-Smirnov Test and all of them were normally distributed, so parametric tests could be used.

Also, Valentín Hernández-Barrera, an expert statistician of Preventive Medicine and Public Health Teaching and Research Unit, Health Sciences Faculty, Universidad Rey Juan Carlos, Madrid, Spain (ORCID: 0000-0001-5790-1959; E-mail: valentin.hernandez@urjc.es) confirms that the statistical tests used are correct and appropriate. Several previous studies, with similar designs to ours, have used the same statistical methods applied in our investigation:

• Kim C, Sohn J-H, Jang MU, Kim S-H, Choi M-G, Ryu O-H, et al. (2015) Association between Visit-to-Visit Glucose Variability and Cognitive Function in Aged Type 2 Diabetic Patients: A Cross-Sectional Study. PLoS ONE 10(7): e0132118.doi:10.1371/journal.pone.0132118

• Wang A, Liu X, Xu J, Han X, Su Z, Chen S, Zhang N, Wu S, Wang Y, Wang Y. Visit-to-Visit Variability of Fasting Plasma Glucose and the Risk of Cardiovascular Disease and All-Cause Mortality in the General Population. J Am Heart Assoc. 2017 Nov 29;6(12):e006757. doi: 10.1161/JAHA.117.006757. PMID: 29187392; PMCID: PMC5779006.

• Xu D, Fang H, Xu W, Yan Y, Liu Y, Yao B. Fasting plasma glucose variability and all-cause mortality among type 2 diabetes patients: a dynamic cohort study in Shanghai, China. Sci Rep. 2016 Dec 22;6:39633. doi: 10.1038/srep39633. PMID: 28004765; PMCID: PMC5177938.

Lastly, Hans DeVries J (Academic Medical Center at the University of Amsterdam, Amsterdam, the Netherlands) in his article entitled “Glucose Variability: Where It Is Important and How to Measure It” (Diabetes. 2013 May;62(5):1405-8. doi: 10.2337/db12-1610. PMID: 23613566), says: “When diabetes investigators want to assess glucose variability, I would recommend coefficient of variation and mean absolute glucose (MAG)”.

- Check for the grammar and typo error.

An expert native translator has reviewed the manuscript and confirms that there are no grammatical errors. Please find enclosed a certificate from the translator.

-Please report the other medical underlyig conditions and how they affect the finding?

Our study has incorporated the main variables that can affect glycemic variability in the multivariate analysis. Study's covariates are as follows: Age, sex, cardiovascular disease, hypertension, use of aspirin, statins, GLP-1 receptor agonists, SGLT-2 inhibitors and DPP-4 inhibitors, baseline FPG and average HbA1c (when at least two measurements)

The studies listed below have used the same or similar control variables as our study.

• Cardoso CRL, Leite NC, Moram CBM, Salles GF. Long-term visit-to-visit glycemic variability as predictor of micro- and macrovascular complications in patients with type 2 diabetes: The Rio de Janeiro Type 2 Diabetes Cohort Study. Cardiovasc Diabetol. 2018 Feb 24;17(1):33. doi: 10.1186/s12933-018-0677-0. PMID: 29477146; PMCID: PMC6389075.

• Orsi E, Solini A, Bonora E, Fondelli C, Trevisan R, Vedovato M, Cavalot F, Gruden G, Morano S, Nicolucci A, Penno G, Pugliese G; Renal Insufficiency and Cardiovascular Events (RIACE) Study Group. Haemoglobin A1c variability is a strong, independent predictor of all-cause mortality in patients with type 2 diabetes. Diabetes Obes Metab. 2018 Aug;20(8):1885-1893. doi: 10.1111/dom.13306. Epub 2018 Apr 19. PMID: 29582548.

An alternative option would have been to use propensity score matching, but this was discarded for two reasons:

1) No studies have ever used the propensity score to measure the effect of glycemic variability on mortality as can be demonstrated by the following search strategy in PubMed: "glycemic variability" AND "propensity score matching": 0 Results..

2) In the limitations section, we have added the following explanation about the not use of propensity score matching.

“Given the observational nature of the present study, individuals with higher GV and lower GV were dissimilar. Therefore, to obtain an accurate picture of the association between GV and all-cause mortality, it was necessary to adjust for differences in both groups in the multivariate analysis. Propensity score matching (PSM) would be a more appropriate alternative that would yield less biased results than standard methods such as Cox regression. However, one of the drawbacks of PSM is the loss of sample in terms of size. In addition, PSM should not be used in practice because our sample size was insufficiently large. Given that propensity scores can only control for observed confounders, they cannot be counted upon to balance unobserved covariates”

Reviewer #2: The current study “Glycemic variability and all-cause mortality in a large prospective southern European cohort of patients with differences in glycemic status” contributes in highlighting the significance to include glycemic variability (GV) as one of the factors to monitor the prognosis of different glycemic status patients.

The finding do suggest how GV could behave as a long-term predictor of all-cause mortality in individuals with certain underlying conditions with variation included .It is rightly understood, with the limitation of the method to satisfy all the variables not limited to duration of the disease, diabetes treatments, or microalbuminuria etc, which would prevent this to be used with confidence.

But the authors do make a valid case for inclusion of GV analysis through their proposed methods so as to derive information for managing the outcome of the diabetic condition of the patients.

The methodological approach to studying the effect of glycemic variability on mortality is similar to that proposed by all the studies published to date. The ideal would have been to ensure that both groups: those exposed to greater glycemic variability and those exposed to lesser glycemic variability, were similar in all the variables predictive of mortality. This approach would have needed to perform a propensity score matching, which was discarded because it would have meant a considerable reduction in the study sample. To date, no article like ours has done so.

There are however, some concerns with the current draft of the manuscript. The authors should follow a consistent mode of representation. E.g. at some places Fig. is used while at some places Figure is written. Kindly adhere to one form of writing.

Thank you for this comment. Sorry for the lack of consistency. Following your suggestion, the text has been completely reviewed and all these errors edited.

---

## [Decision Letter · Decision Letter 1]

6 Jul 2022

Glycemic variability and all-cause mortality in a large prospective southern European cohort of patients with differences in glycemic status.

PONE-D-21-35463R1

Dear Dr.Salinero-Fort,

We’re pleased to inform you that your manuscript has been judged scientifically suitable for publication and will be formally accepted for publication once it meets all outstanding technical requirements.

Kind regards,

Venkata Naga Srikanth Garikipati, PhD

Academic Editor

PLOS ONE

Additional Editor Comments (optional):

Reviewers' comments:

Reviewer's Responses to Questions

**Comments to the Author**

1. If the authors have adequately addressed your comments raised in a previous round of review and you feel that this manuscript is now acceptable for publication, you may indicate that here to bypass the “Comments to the Author” section, enter your conflict of interest statement in the “Confidential to Editor” section, and submit your "Accept" recommendation.

Reviewer #2: All comments have been addressed

Reviewer #3: All comments have been addressed

2. Is the manuscript technically sound, and do the data support the conclusions?

Reviewer #2: Yes

Reviewer #3: Yes

3. Has the statistical analysis been performed appropriately and rigorously? 

Reviewer #2: Yes

Reviewer #3: Yes

4. Have the authors made all data underlying the findings in their manuscript fully available?

Reviewer #2: Yes

Reviewer #3: Yes

5. Is the manuscript presented in an intelligible fashion and written in standard English?

Reviewer #2: Yes

Reviewer #3: Yes

6. Review Comments to the Author

Reviewer #2: (No Response)

Reviewer #3: (No Response)

7. PLOS authors have the option to publish the peer review history of their article (what does this mean?). If published, this will include your full peer review and any attached files.

Reviewer #2: **Yes: **Manju Narwal

Reviewer #3: No

---

## [Editor Report · Acceptance letter]

15 Jul 2022

PONE-D-21-35463R1 

Glycemic variability and all-cause mortality in a large prospective southern European cohort of patients with differences in glycemic status. 

Dear Dr. Salinero-Fort:

I'm pleased to inform you that your manuscript has been deemed suitable for publication in PLOS ONE. Congratulations! Your manuscript is now with our production department. 

Kind regards, 

on behalf of

Dr. Venkata Naga Srikanth Garikipati 

Academic Editor

PLOS ONE